# Distinct subdivisions of human medial parietal cortex support recollection of people and places

Edward H Silson[1†]*, Adam Steel[1,2†], Alexis Kidder[1], Adrian W Gilmore[1], Chris I Baker[1]

[1]Laboratory of Brain & Cognition, National Institute of Mental Health, Bethesda, United States; [2]Wellcome Centre for Integrative Neuroimaging, FMRIB, Nuffield Department of Clinical Neurosciences, University of Oxford, Oxford, United Kingdom

**Abstract** Human medial parietal cortex (MPC) is implicated in multiple cognitive processes including memory recall, visual scene processing and navigation, and is a core component of the default mode network. Here, we demonstrate distinct subdivisions of MPC that are selectively recruited during memory recall of either specific people or places. First, distinct regions of MPC exhibited differential functional connectivity with medial and lateral regions of ventral temporal cortex (VTC). Second, these same medial regions showed selective, but negative, responses to the visual presentation of different stimulus categories, with clear preferences for scenes and faces. Finally, and most critically, these regions were differentially recruited during memory recall of either people or places with a strong familiarity advantage. Taken together, these data suggest that the organizing principle defining the medial-lateral axis of VTC is reflected in MPC, but in the context of memory recall.
DOI: https://doi.org/10.7554/eLife.47391.001

*For correspondence:
ed.silson@nih.gov

†These authors contributed equally to this work

Competing interests: The authors declare that no competing interests exist.

## Introduction

Human medial parietal cortex (MPC), a core component of the default mode network (DMN) (*Andrews-Hanna et al., 2010*), comprises a relatively large expanse of cortex, spanning the parieto-occipital sulcus to the splenium of the corpus collosum anteriorly and dorsally to include the precuneus and both the ventral and dorsal portions of the posterior cingulate cortex (*Bzdok et al., 2015*). MPC is associated with a diverse set of cognitive functions, including (but not restricted to) memory recall (*Vilberg and Rugg, 2008*; *Wagner et al., 2005*; *Gilmore et al., 2015*; *Kim, 2013*) visual scene perception (*Epstein et al., 2007*; *Silson et al., 2016*; *Baldassano et al., 2013*), scene construction (*Hassabis et al., 2007*), processing of spatial and other contextual associations (*Bar and Aminoff, 2003*), navigation (*Epstein, 2008*), future thinking (*Benoit and Schacter, 2015*; *Szpunar et al., 2007*; *Gilmore et al., 2016*), and mental orientation (*Peer et al., 2015*). Given such diverse recruitment of MPC across cognitive domains historically considered largely independent (e.g. visual processing, memory), the absence of a clear consensus with regard to the function and overarching organization of MPC is perhaps unsurprising (*Gilmore et al., 2015*; *Chrastil, 2018*).

Network analyses using resting-state-functional-connectivity (RSFC) have identified either a single DMN 'hub' region (Buckner et al. 2008) or multiple networks (*Power et al., 2011*; *Braga and Buckner, 2017*; *Gilmore et al., 2018*) that are often described as DMN subnetworks (*Andrews-Hanna et al., 2010*; *Yeo et al., 2011*; *Doucet et al., 2011*). Task-based analyses also suggest a fractionation beyond a single region (*Andrews-Hanna et al., 2010*; *Peer et al., 2015*; *Chrastil, 2018*; *Nelson et al., 2012*) with, for example, evidence that the ventral and dorsal portions of posterior

cingulate cortex interact differently with the rest of DMN during cognitive control (*Leech et al., 2011*). Further, recent work has reframed MPC (and the DMN) in terms of large-scale cortical gradients (*Margulies et al., 2016*), conceptualizing MPC as the most abstract extension of the ventral visual pathway (*Murphy et al., 2018*; *Murphy et al., 2019*). Beyond MPC's link with the DMN, others have described divisions of MPC along both the posterior-anterior and ventral-dorsal axes in terms of cytoarchitecture (*Vogt, 2009*), structural connectivity (*Parvizi et al., 2006*), RSFC (*Margulies et al., 2009*; *Vidaurre et al., 2018*; *Bzdok et al., 2015*), and electrocorticography (*Foster and Parvizi, 2012*; *Daitch and Parvizi, 2018*).

The question of the underlying functional organization of MPC is clearly complicated and could potentially benefit from a simple and more unified perspective. A promising lead on one such organization has come from recent work demonstrating a strong functional link between anterior ventral temporal cortex (VTC) and MPC (*Baldassano et al., 2013*; *Baldassano et al., 2016*; *Silson et al., 2016*). Specifically, a small region of MPC directly anterior of (visually) scene-selective medial place area (MPA) (*Silson et al., 2016*) showed strong functional connectivity with anterior portions of scene-selective parahippocampal place area (*Epstein, 2008*) (aPPA), located in medial VTC. This connectivity-defined region overlaps with regions of MPC engaged during memory recall (*Silson et al., 2019*), suggesting that the ventral/posterior aspect of MPC may contain distinct areas biased toward scene processing for vision and memory, respectively.

What else might we learn from examining potential links between MPA and VTC? Whilst the previous functional link between MPC and VTC was based upon parcellating PPA along its posterior-anterior axis, the functional organization of VTC varies more dramatically along the orthogonal medial-lateral axis. Indeed, multiple functional dimensions are thought to be represented along this axis, including category preference (*Kanwisher et al., 1997*; *Deen et al., 2017*) (e.g. scenes, objects, tools and faces), eccentricity (*Levy et al., 2001*; *Arcaro and Livingstone, 2017*) (e.g. peripheral, foveal), animacy (*Konkle and Caramazza, 2013*), and even real-world size (*Konkle and Oliva, 2012*). Further, the mid-fusiform sulcus (MFS) (*Weiner et al., 2014*) has been identified as an anatomical landmark marking a transition point within each dimension (e.g. scene-selectivity medial of the MFS, face-selectivity lateral of the MFS). These robust functional differences across the medial-lateral axis of VTC—and their well-characterized, category-specific nature—may provide an effective perspective from which to investigate the organizational structure of MPC.

To investigate whether the functional organization of MPC reflects that of VTC, we conducted three independent fMRI experiments. First, we found that distinct subdivisions of MPC have preferential functional connectivity to anterior portions of medial and lateral VTC, respectfully. Second, these MPC subdivisions showed differential evoked responses to the presentation of different visual categories, with clear evidence for scene and face preferences. Third, and most critically, these subdivisions were selectively recruited during memory recall of either specific places (i.e. scenes) or specific people (i.e. faces). Finally, an independent whole-brain analysis of memory recall effects revealed an even finer division within MPC, with four identifiable regions showing an alternating (place/people) pattern of selective recruitment during memory recall.

Taken together, these findings provide converging evidence for a reflection of the functional organization of VTC in MPC. This organization was evident at rest, in response to visual stimulation, and most strikingly, during memory recall. The alternating pattern of responses throughout MPC provides a framework for understanding the broader functional organization of MPC and may tie together many of the disparate observations reported across the literature. Collectively, these data support the notion that the functional organization defining the medial-lateral axis of VTC is reflected along the ventral/posterior-dorsal/anterior axis MPC, but in the context of memory retrieval.

## Results

### Subdivisions of MPC show preferential functional connectivity with medial and lateral portions of VTC

To determine whether the functional organization along the medial-lateral axis of VTC is reflected in MPC, we first utilized resting-state functional connectivity data (n = 65). Six regions of interest (ROIs) were defined anatomically in each hemisphere that divided VTC along both the posterior-anterior

and medial-lateral axes with respect to the MFS (*Weiner et al., 2014*), allowing us to characterize the connectivity profile between VTC and MPC more precisely (*Figure 1a*). A winner-take-all analysis (see Materials and methods) revealed a ventral-posterior MPC region (referred to as MPC ventral, MPCv) that showed strongest connectivity with the anterior medial ROI and an adjacent dorsal-anterior region (referred to as MPC dorsal, MPCd) that showed strongest connectivity to the anterior lateral ROI (*Figure 1b*). Such a pattern of connectivity suggests a reflection of the functional organization defining the medial-lateral axis of VTC along the posterior/ventral-anterior/dorsal axis of MPC.

### Subdivisions of MPC show differential responses to visually presented categories

Having identified distinct subdivisions of MPC based on differential functional connectivity with anterior portions of medial and lateral VTC, we next sought to determine whether these subdivisions would respond differentially to the visual presentation of different stimulus categories (Scenes, Faces, Buildings, Bodies, Objects and Scrambled Objects) (see Materials and methods). Differentiation on the basis of stimulus category would be reminiscent of the category-preference changes along the medial-lateral axis of VTC.

In a second independent group of participants (n = 29), we calculated the mean response to each category (given as the *t*-value versus baseline) in both ROIs and hemispheres separately. Unlike category-selective regions of VTC (e.g. PPA, Fusiform Face Area, FFA), which typically exhibit positive responses to the presentation of visual stimuli, we observed negative response magnitudes to all

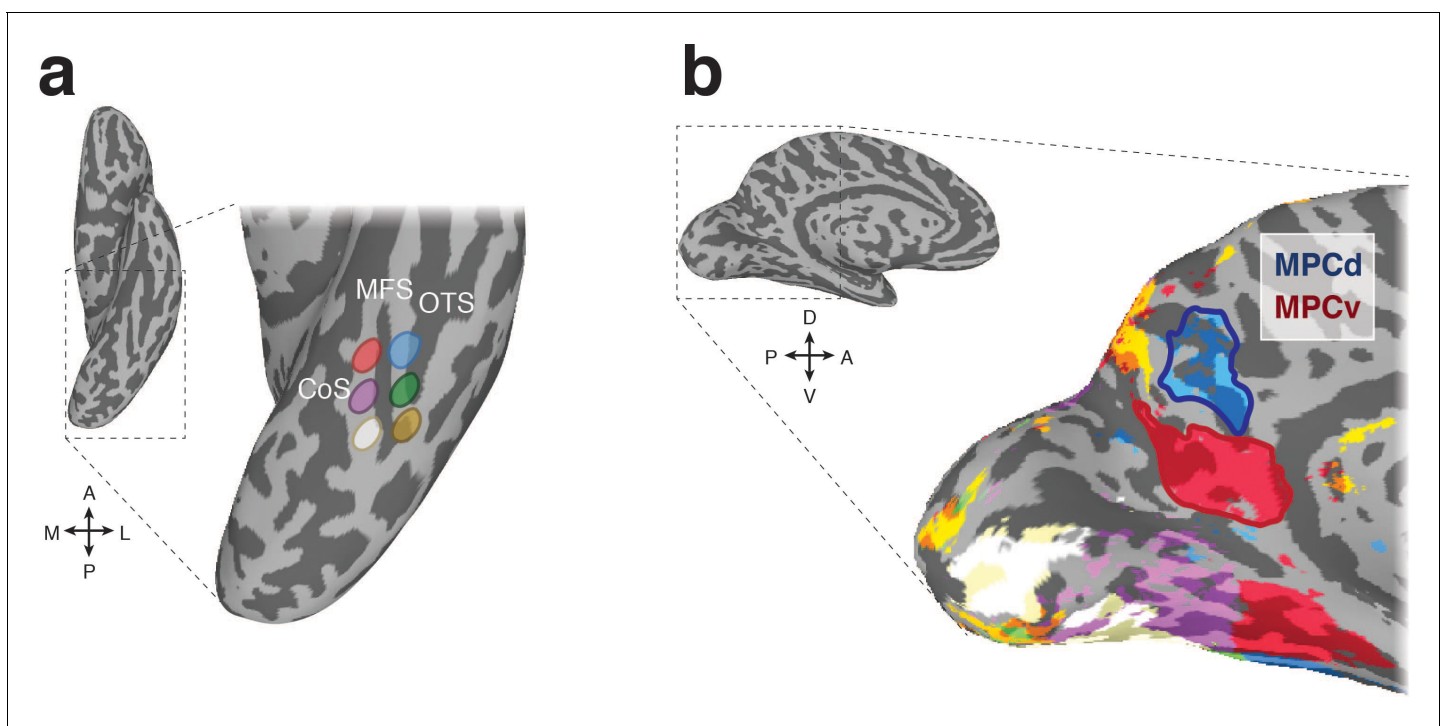

**Figure 1.** Resting-state functional connectivity seed regions and connectivity-defined regions of interest. (**a**) A ventral view of the left hemisphere is shown with the ventral temporal cortex (VTC) highlighted with the dashed-black box, which is enlarged inset. Overlaid onto this enlarged surface are the six anatomically defined regions of interest that divide VTC along both the posterior-anterior and medial-lateral axes with respect to the mid-fusiform sulcus (MFS). The occipitotemporal sulcus (OTC) and collateral sulcus (CoS) are also labeled for reference. (**b**) A medial view of the left hemisphere is shown with medial parietal cortex (MPC) highlighted by the dashed-black box, which is enlarged inset. Overlaid onto this enlarged surface is the result of the winner-take-all functional connectivity analysis. Colors on the brain correspond to the color of the anatomical ROIs in a. Within MPC, two separate regions are clearly visible. The ventral/posterior region (red-outline) is preferentially connected to anterior medial portions of VTC, whereas the dorsal/anterior region (blue-outline) is preferentially connected to anterior lateral portions of VTC. We define these resting-state ROIs as MPC ventral (MPCv) and MPC dorsal (MPCd), respectively.

DOI: https://doi.org/10.7554/eLife.47391.002

categories within both MPC subdivisions. Despite this general tendency for negative magnitudes, responses also appeared to differentiate on the basis of category, with scenes evoking the strongest response (i.e. least negative) in MPCv and faces evoking the strongest response in MPCd of both hemispheres (*Figure 2*).

To explore these effects further, mean response magnitudes for each category were subjected to a one-way repeated measures Analysis of Variance (ANOVA) with Category (six levels) as a within-participant factor. MPCv exhibited a significant main effect of Category in both the left ($F_{(5, 140)}$=71.38, $p<0.0001$, partial $\eta^2 = 0.72$) and right ($F_{(5, 140)}$=49.46, $p<0.0001$, partial $\eta^2 = 0.64$)

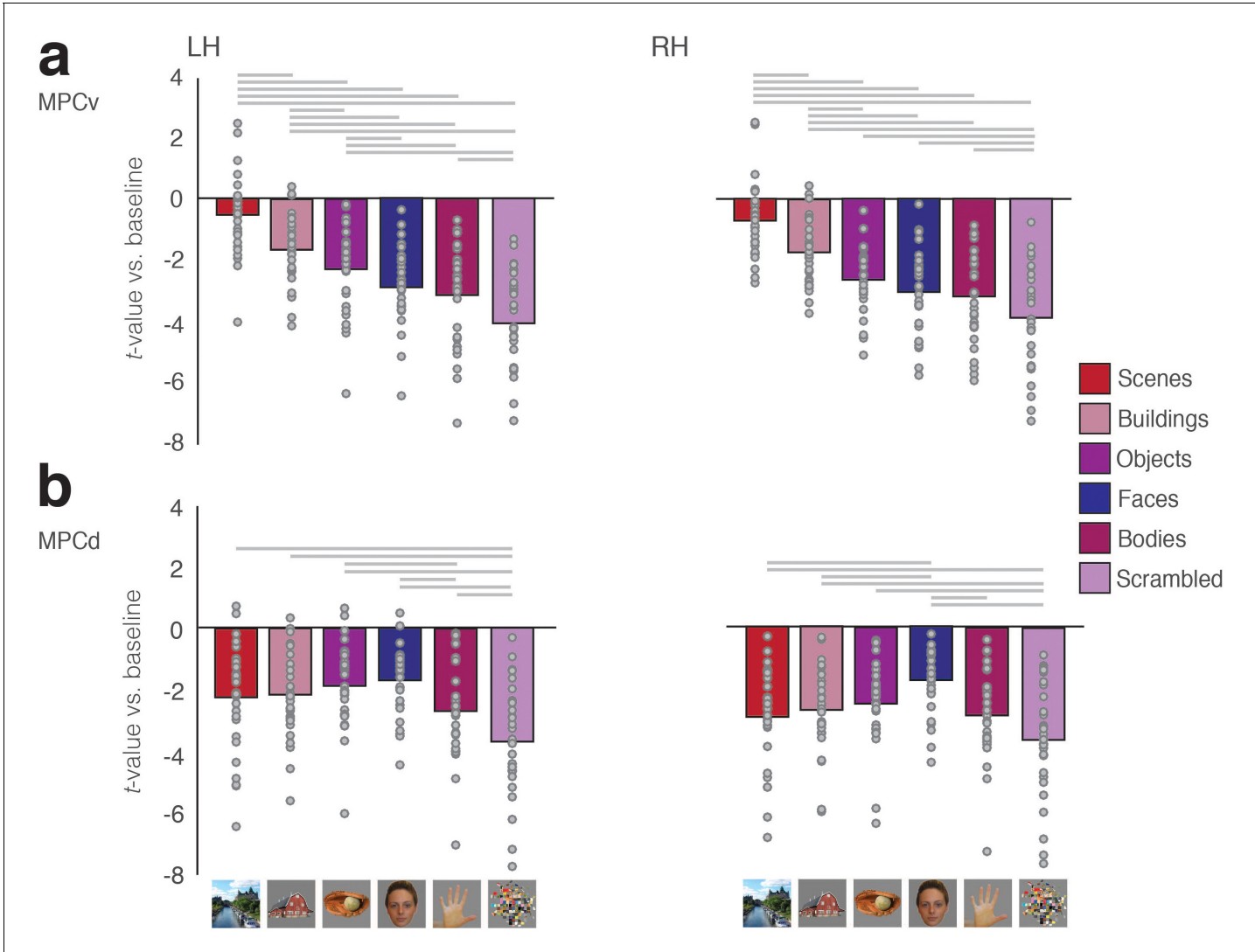

**Figure 2.** Negative responses in MPC to visually presented stimuli. (a) Bars represent the mean response magnitude (given by the *t*-value versus baseline) for all six stimulus categories in MPCv of both hemispheres. These responses have been rank ordered from strongest (i.e. closest to baseline) to weakest. Single participant data points are shown for each category. Gray lines depict pairwise-comparisons that survived Bonferroni correction (p<0.01). The response to scenes was significantly different compared to all other categories in both hemispheres. (b) Bars represent the mean response magnitude for all six stimulus categories in MPCd of both hemispheres. Bars are in the same order as in a, to highlight the different response profiles. Gray lines depict pairwise-comparisons that survived Bonferroni correction (p<0.01). The response to faces was significantly different from the majority of the other categories.

DOI: https://doi.org/10.7554/eLife.47391.003

The following source data is available for figure 2:

**Source data 1.** Visually evoked responses in MPCv/MPCd.

DOI: https://doi.org/10.7554/eLife.47391.004

hemispheres. Consistent with the stronger functional connectivity with medial VTC, the response to scenes was significantly different from other stimulus categories in both hemispheres ($t > 5.34$, p<0.001, in all cases, Bonferroni corrected) (*Figure 2a*).

MPCd also exhibited a significant effect of Category within both left ($F_{(5, 140)}$=19.28, p<0.0001, partial $\eta^2$ = 0.48) and right $F_{(5, 140)}$=12.66, p<0.0001, partial $\eta^2$ = 0.31) hemispheres. However, in this case the response to faces was only significantly different from Buildings and Scrambled Objects in the left hemisphere ($t > 3.86$, p<0.001, Bonferroni corrected; p>0.05, in all other cases), whereas the response to faces was significantly different from all categories except Objects (p>0.05) in the right hemisphere ($t > 3.54$, p<0.001, Bonferroni corrected) (*Figure 2b*). Collectively these results demonstrate a preference for scenes and faces within MPCv and MPCd, respectively. The overall pattern of negative responses evoked by visual stimuli is consistent with the widely-reported negative responses within MPC (and the broader DMN) when orienting to external stimuli[27]. However, motivated by the apparent scene and face preference within these regions and the fact that MPC is typically engaged positively during introspective tasks such as scene-construction from memory and mental imagery, we hypothesized that these MPC subdivisions would become differentially recruited during memory recall of either specific places (MPCv) or specific people (MPCd), respectively.

## Subdivisions of MPC differentially recruited during memory recall of specific places or specific people

To investigate the hypothesis that MPCv/MPCd would be differentially recruited during memory recall of specific places and people, respectively, we conducted a memory recall experiment in a third independent group of participants (n = 24). Participants performed six runs of a memory recall task, in which they were cued to recall from memory either specific places or specific people. Here, a simple 2 × 2 design was employed with two categories (Places, People) and two levels of familiarity (Famous, Personal) (*Figure 3a*). Such a design allowed us to test the predictions that MPCv/MPCd would be selectively recruited during recall of specific places and people, respectively, whilst the addition of personally relevant stimuli provided a means to assess whether such category-selective recruitment was dependent on the richness of internal representations. In order to test this hypothesis, we looked in each region for the effects of category, familiarity and their potential interaction.

Subdivisions within MPC showed strikingly different response profiles during memory recall (derived by averaging the evoked responses across all trials per condition). Within MPCv, responses were maximally positive (relative to baseline) for the recall of personally familiar places, whereas responses during recall of famous people were maximally negative (*Figure 3b*). In contrast, responses within MPCd were maximally positive for the recall of personally familiar people and maximally negative during recall of famous places (*Figure 3c*). To quantify these responses, we calculated the mean contrast response (given by the *t*-value versus baseline) within each ROI to all conditions from the GLM analysis (See Materials and methods). These responses were then subjected to a three-way repeated measures ANOVA for each ROI separately, with Category (People, Places), Familiarity (Famous, Personal) and Hemisphere (Left, Right) as within-participant fact.

## MPCv selectively recruited during memory recall of specific places

Within MPCv, the main effects of Category ($F_{(1, 23)}$=75.40 p=$1.02^{-8}$, partial $\eta^2$ = 0.76), Familiarity ($F_{(1, 23)}$=128.61 p=$6.78^{-11}$, partial $\eta^2$ = 0.85) and Hemisphere ($F_{(1, 23)}$=4.92 p=0.03, partial $\eta^2$ = 0.17) were significant, reflecting on average greater responses for the recall of places over people, personal over famous stimuli and in the right compared to left hemisphere, respectively. However, these main effects were qualified by a significant three-way interaction (Category by Familiarity by Hemisphere: $F_{(1, 23)}$=7.19 p=0.01, partial $\eta^2$ = 0.24). This interaction reflects a larger familiarity difference (Personal >Famous) between categories (Place >People), in the right over left hemisphere. Further, we performed separate two two-way ANOVAs in each hemisphere separately with Category and Familiarity as factors. In both hemispheres, the Category by Familiarity interaction was significant (Left: $F_{(1, 23)}$=31.19, p=0.00001, partial $\eta^2$ = 0.56; Right: $F_{(1, 23)}$=49.51, p=$3.59^{-7}$, partial $\eta^2$ = 0.68), reflecting a larger familiarity difference for the recall of places over people in both hemispheres (*Figure 4a*).

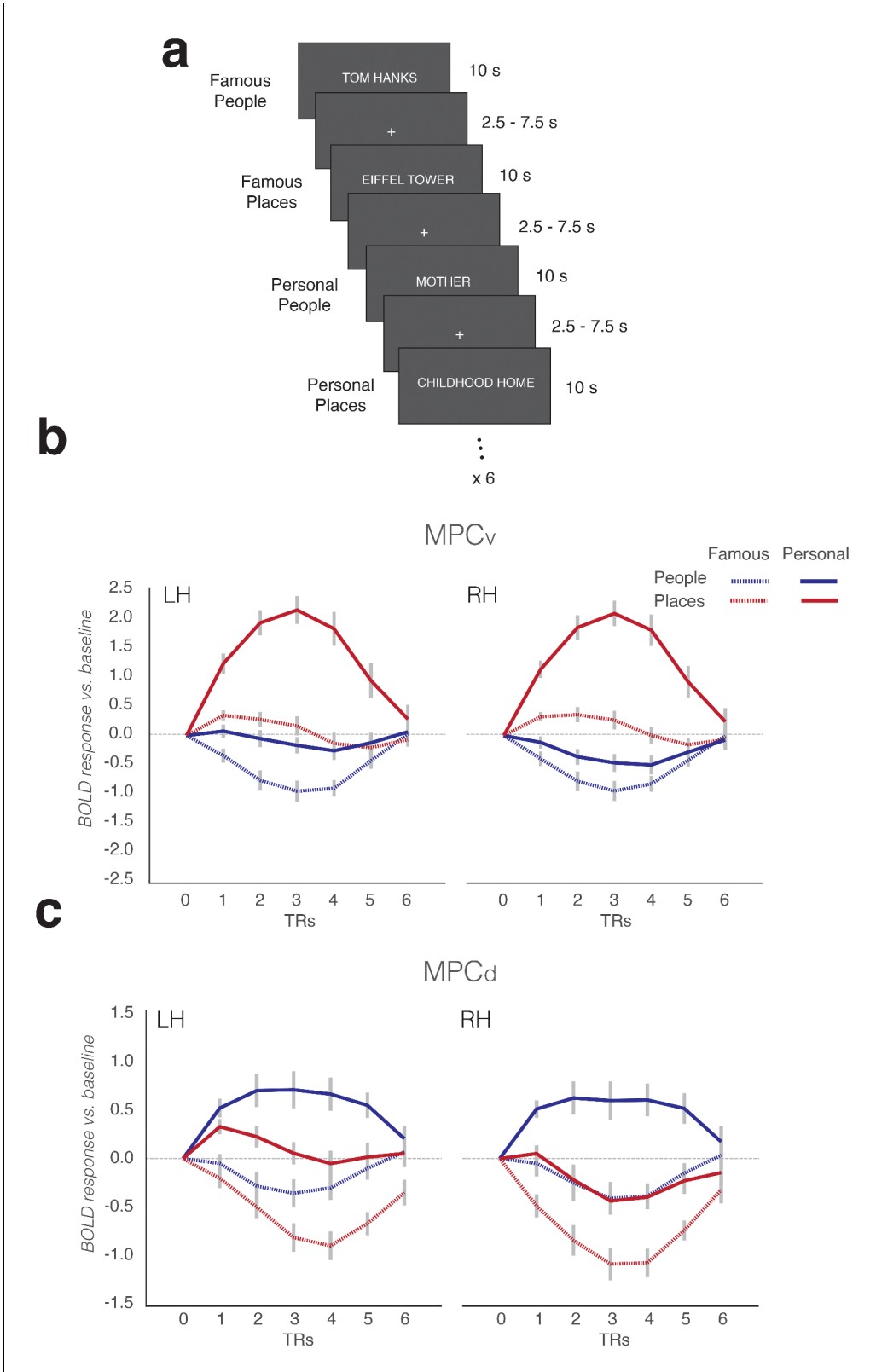

**Figure 3.** Memory task schematic and average BOLD responses to all conditions in MPC subdivisions. (a) During the memory task, participants were given trial-wise instructions to recall from memory either famous people, famous places, personally familiar people or personally familiar places, respectively. Participants were asked to visualize the trial target from memory as vividly as possible for the duration of the trial (10 s). Trials were separated by a variable inter-trial-interval (2.5–7.5 s). Participants completed 6 runs of the memory task. Each run contained six randomized trials from
*Figure 3 continued on next page*

*Figure 3 continued*

each condition. (b) Average response curves from left and right MPCv (relative to baseline) are shown for all conditions in both hemispheres. Response curves were generated by first measuring the average response across the ROI for each trial (6 TR's from trial onset) and then averaging across trails of the same condition. These responses were then averaged across participants and plotted for each condition separately (Famous people – dashed blue, Famous places – dashed red, Personal people – solid blue and Personal places – solid red). MPCv is maximally recruited during the recall of personal places. The patterns of response are very similar across hemispheres. (c) Same as b, but for MPCd. In contrast to b, MPCd is maximally recruited during the recall of personal people. Again, this pattern is consistent across hemispheres. Response curves were normalized to begin at baseline (zero) for each trial separately. Gray-lines represent the standard error of the mean (sem) across participants for each condition and TR.

DOI: https://doi.org/10.7554/eLife.47391.005

## MPCd selectively recruited during memory recall of specific people

Within MPCd, the main effects of Category ($F_{(1, 23)}$=47.53 p=4.98$^{-7}$, partial $\eta^2$ = 0.67), Familiarity ($F_{(1, 23)}$=82.33 p=4.62$^{-9}$, partial $\eta^2$ = 0.78) and Hemisphere ($F_{(1, 23)}$=10.70 p=0.003, partial $\eta^2$ = 0.32) were again significant, reflecting on average greater responses for the recall of people over places, personal over famous stimuli and in the right compared to left hemisphere, respectively. Whilst, we did not observe a significant three-way interaction, several significant two-way interactions were observed. Importantly, the Category by Familiarity interaction ($F_{(1, 23)}$=7.89 p=0.01, partial $\eta^2$ = 0.25), was significant, which reflects a larger familiarity difference for the recall of people over places with no clear difference between hemispheres (*Figure 4b*). (see *Supplementary file 1b* for full statistical breakdown).

## Consistent topography of memory recall effects within MPC

Both MPC subdivisions showed differential recall effects for places and people, respectively, coupled with an overall familiarity advantage. The topography of this differential recruitment during recall was strikingly consistent across individuals. Indeed, throughout MPC comparisons of the peak locations for the recall of personal places and personal people demonstrates a consistent shift along the ventral/posterior– dorsal/anterior axis. Across all participants and hemispheres, the peak response during recall of personal people was always anterior and dorsal to the peak response during recall of personal places (*Figure 5*).

## Alternating pattern of place and people memory recall throughout MPC

Having established that subdivisions of MPC are differentially recruited during memory recall, we next sought to determine whether areas outside of these initial ROIs showed similar effects. Accordingly, we performed a whole-brain Linear-Mixed-Effects (LME) modelling analysis to look for regions of the brain displaying main effects of Category (Places, People), Familiarity (Famous, Personal) and their interaction (see Materials and methods). At the whole-brain level, we did not observe any regions showing a significant interaction (at the selected statistical threshold), although significant responses to both main effects were present. The main effect of Category (collapsed across familiarity) revealed a complex pattern of differential recruitment throughout the brain. Most strikingly, along the ventral/posterior-dorsal/anterior axis of MPC, we observed an alternating pattern of memory recall: four adjacent subdivisions that alternated between being selectively recruited by the recall of places, then people, then places and finally people (*Figure 6a*). Notably, the first two subdivisions (refereed to here as ROIs 1 and 2) were largely equivalent to the connectivity-defined MPCv and MPCd (see Supplementary Material for spatial overlap). Thus, this analysis not only confirmed the differential recruitment during memory recall of MPCv and MPCd, but also, revealed two anterior subdivisions within bilateral MPC that showed similar patterns of selective recruitment (ROIs 3 and 4). Strong memory recall for places was also present in aPPA in both hemispheres (*Figure 6a*). In contrast to the alternating pattern of place and people recall, the effect of familiarity (collapsed across category) manifested as an advantage for the recall of personally familiar over famous stimuli, irrespective of category within a relatively large swath of MPC (*Figure 6b*).

To quantify the selective recruitment within each of these four MPC subdivisions in an independent manner, we implemented a split-half analysis. First, in each participant, we divided the six memory runs into odd and even datasets (three runs each). Next, we performed the same LME

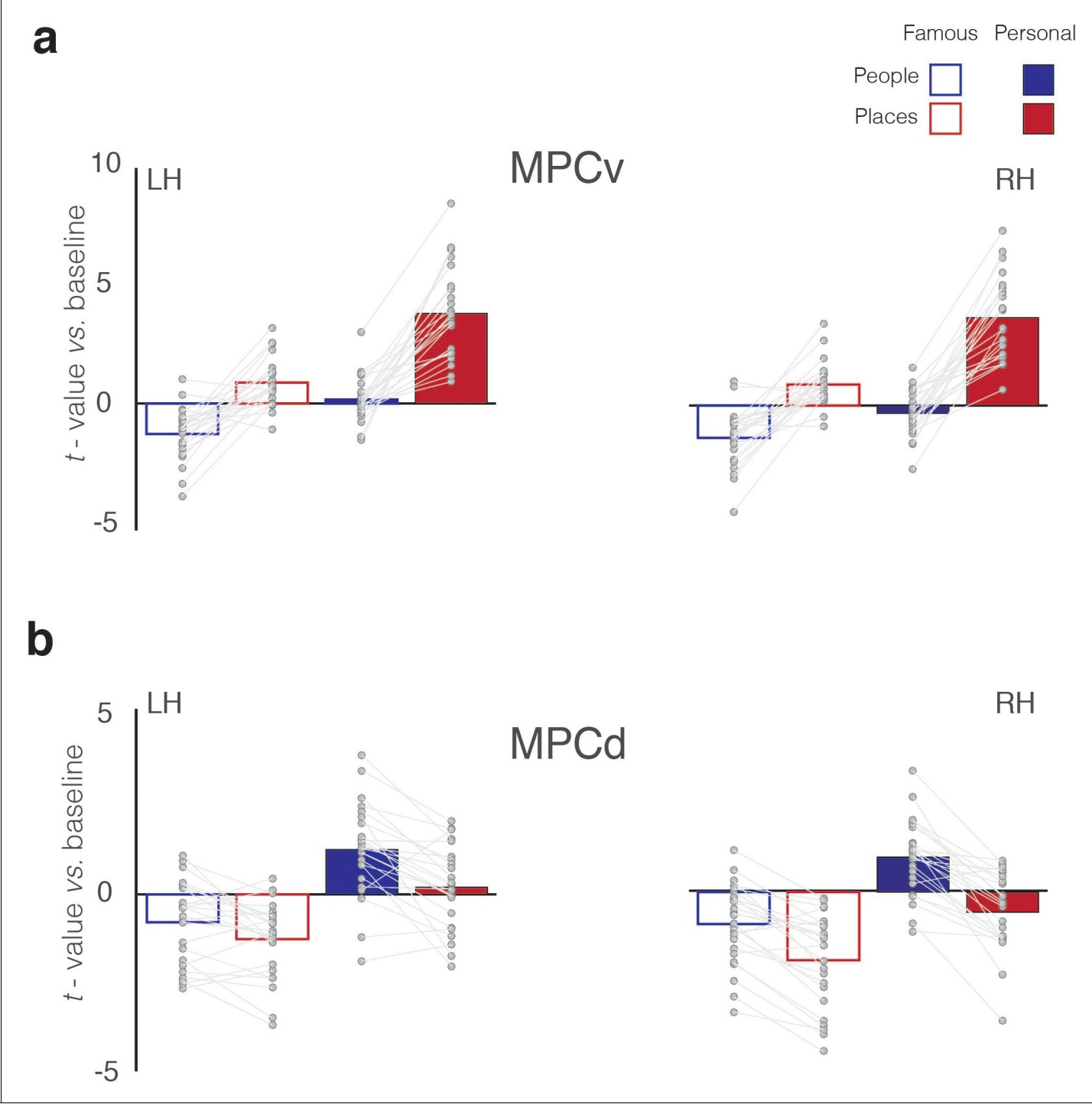

**Figure 4.** Magnitude of memory recall for all conditions in MPCv and MPCd. (a) Bars represent the mean response magnitude for each condition (*t*-value versus baseline) in MPCv of both hemispheres (Famous people – blue open bars, Famous places – red open bars, Personal people – blue closed bars, Personal places – red closed bars). Data points for each participant are connected. In both hemispheres, MPCv is positively recruited during the recall of famous places and personal places, whereas responses during the recall of people (either famous or personal) are largely negative, reflecting a Category preference for places. MPCv also exhibits a familiarity effect and is maximally recruited during the recall of personal places, reflecting the effect of Familiarity. The interaction between Category and Familiarity is also evident. Indeed, there is a larger category difference (places-people) in the personal over famous conditions. (b) Bars represent the mean response magnitude for each condition in MPCd of both hemispheres. Here, MPCd is only positively recruited during recall of personal people, reflecting both a Category preference for people and a Familiarity effect. The interaction between Category and Familiarity is also evident: there is a larger category difference (places-people) in the personal over famous conditions.

*Figure 4 continued on next page*

*Figure 4 continued*

DOI: https://doi.org/10.7554/eLife.47391.006

The following source data is available for figure 4:

**Source data 1.** Memory recall effects in MPCv/MPCd.

DOI: https://doi.org/10.7554/eLife.47391.007

analysis as above in each dataset separately (see Materials and methods). Between the odd and even splits, the topography and magnitude of the effect of category was robust and highly correlated across splits and hemispheres, respectively (left hemisphere: r = 0.81, $R^2$ = 0.65; right hemisphere: r = 0.84, $R^2$ = 0.71) (*Figure 7a*). In order to determine estimates of effect size, we defined each MPC subdivision in one half the data (e.g. Odd) and sampled the responses to all conditions from the other half (e.g. Even). This process was then reversed, and the average computed. We observed a consistent and alternating pattern of selective recruitment throughout MPC. Recall of specific places selectively recruited ROIs 1 and 3, whereas recall of specific people selectively recruited ROIs 2 and 4. Consistent with our initial analyses, all four MPC subdivisions exhibited a

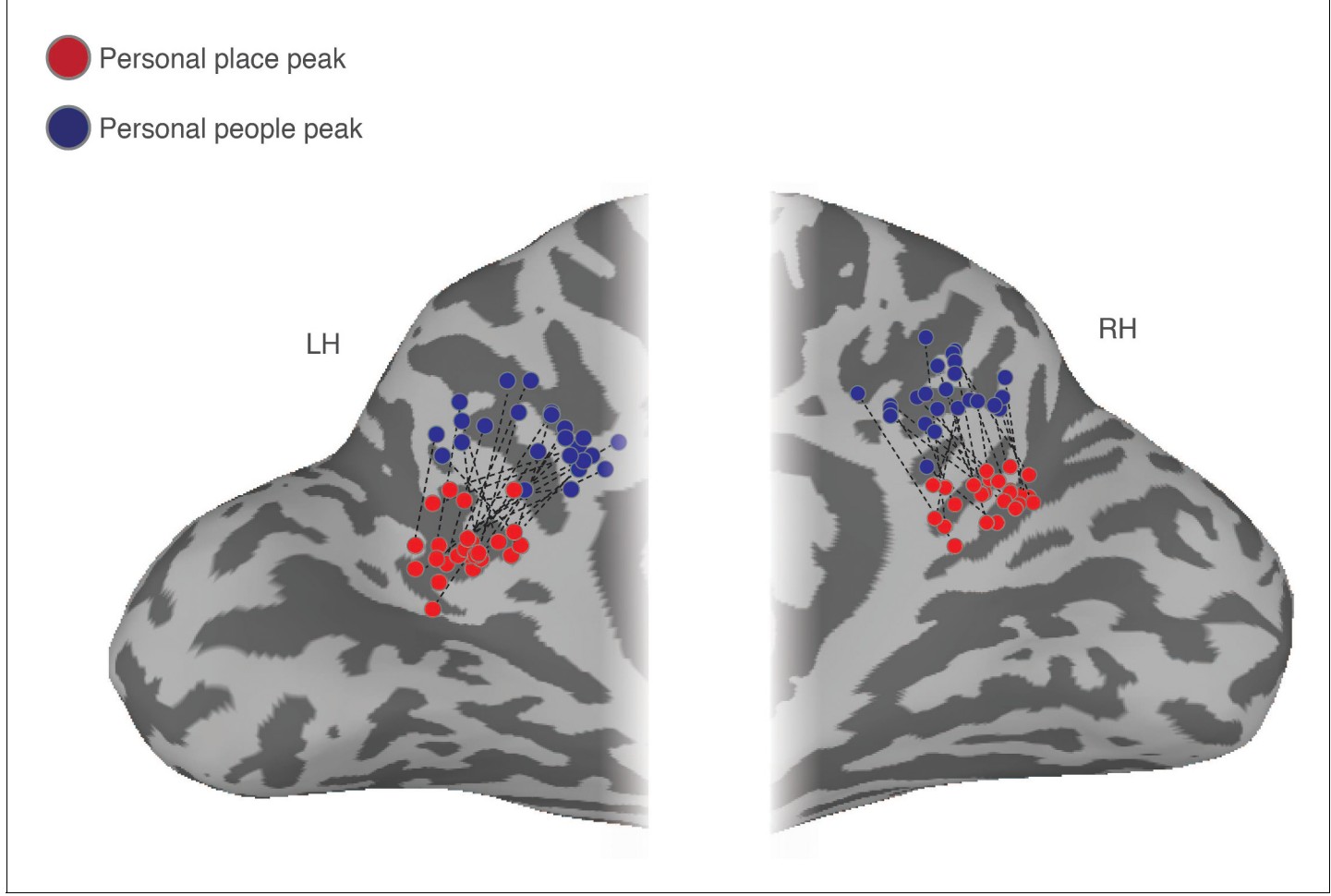

**Figure 5.** Ventral/posterior – dorsal/anterior shift in peak place memory and peak people memory. Enlarged partial views of the posterior medial portion of both the left and right hemispheres are shown. Overlaid onto these enlarged surfaces are the locations of the peak responses during recall of personal places (red dots) and recall of personal people (blue dots) for each participant. The corresponding peaks are connected for each participant with a dashed black line. Across participants, there is a consistent ventral/posterior – dorsal/anterior shift in the peak location of memory recall, such that the peak for recall of personal places is never posterior or ventral of the peak for recall of personal people.

DOI: https://doi.org/10.7554/eLife.47391.008

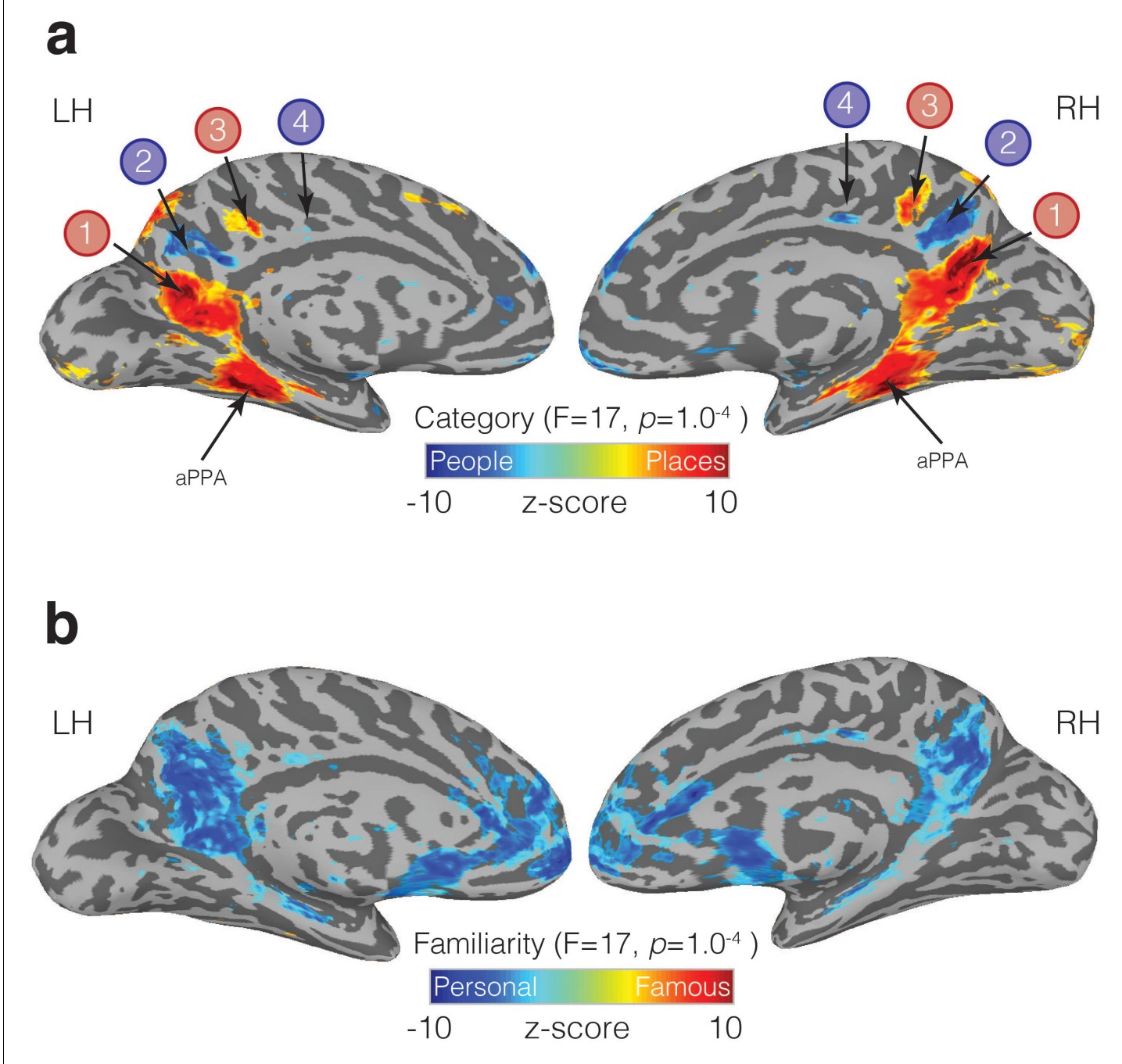

**Figure 6.** Familiarity and Category-selective recall in MPC. (**a**) Medial views of both the left and right hemispheres of a representative participant are shown. Overlaid onto these surfaces is the main effect of Category from the linear-mixed-effects analysis (node-wise $p=1.0^{-4}$, $q = 5.8^{-4}$). Cold colors represent regions of the brain more active during the recall of people (famous and personal), whereas hot colors represent regions of the brain more active during recall of places (famous and personal). An alternating pattern of memory recall is evident within MPC along the ventral/posterior – dorsal/anterior axis. ROIs 1 and 2, correspond largely to our initial resting-state ROIs (MPCv, MPCd), whereas the anterior pair of regions was not defined initially. We also observe significant place recall in aPPA, and some small clusters of significant people recall in anterior cingulate cortex. (**b**) The same medial views are shown but overlaid is the main effect of Familiarity (node-wise $p=1.0^{-4}$, $q = 5.8^{-4}$). Cold colors represent regions of the brain more active during the recall of personally familiar stimuli (places and people), whereas hot colors represent regions of the brain more active during the recall of famous stimuli (places or people). A large swath of MPC exhibits an overwhelming Familiarity effect with greater activity during recall of personal over famous stimuli. Familiarity effects were also present in the anterior cingulate cortex, insula and ventral medial prefrontal cortex.
DOI: https://doi.org/10.7554/eLife.47391.009

The following source data and figure supplements are available for figure 6:

*Figure 6 continued on next page*

*Figure 6 continued*

**Figure supplement 1.** Memory recall effects on the lateral surface.

DOI: https://doi.org/10.7554/eLife.47391.010

**Figure supplement 1—source data 1.** Memory recall effects in PPA and FFA.

**Figure supplement 2.** Memory recall effects in PPA and FFA.

DOI: https://doi.org/10.7554/eLife.47391.012

**Figure supplement 2—source data 1.** Memory recall effects in Hippocampus and Amygdala.

**Figure supplement 3.** Memory recall effects in the Hippocampus and Amygdala.

DOI: https://doi.org/10.7554/eLife.47391.014

familiarity advantage, which manifested as a selective enhancement in response for personally familiar items (Places = ROIs 1 and 3, People = ROIs, 2 and 4) (see *Supplementary file 1c-1f* for full statistical breakdown).

## Memory recall effects beyond MPC

Significant effects of place and people recall were also evident throughout the brain. In particular, the posterior angular gyrus, inferior temporal sulcus, and superior frontal regions were recruited during place recall, whereas the recall of people recruited the insula and anterior temporal regions, particularly in the right hemisphere (*Figure 6—figure supplement 1*). Advantages for the recall of personally familiar stimuli were present within anterior cingulate cortex and insula in both hemispheres (*Figure 6b*), as well as regions on the lateral surface, including superior-frontal, the superior-temporal sulcus, and angular gyrus/caudal inferior parietal lobule (*Figure 6—figure supplement 1*). In contrast, recall effects associated with famous over familiar stimuli were sparse and non-significant.

Memory recall effects were also observed within functionally-defined scene- and face-selective regions of VTC (i.e. PPA, FFA). Both regions were recruited during recall of items from their preferred category (i.e. greater response to place-specific memory in PPA, greater response to people-specific memory in FFA), although the magnitude of these memory effects were markedly weaker than within MPC (*Figure 6—figure supplement 2* and see *Supplementary file 1g* for full statistical breakdown).

In addition to cortical ROIs, significant memory recall effects were also present in the hippocampus and amygdala bilaterally. The hippocampus showed an effect of category, with larger responses during recall of places over people, whilst also showing a strong familiarity advantage with larger responses for personally familiar over famous stimuli (*Figure 6—figure supplement 3*). In contrast, the amygdala showed only an effect of category, with larger responses during recall of people irrespective of the level of familiarity (*Figure 6—figure supplement 3*; see *Supplementary file 1h* for full statistical breakdown).

## Discussion

Across three independent fMRI experiments, we demonstrate two major subdivisions of MPC that exhibit i) differential functional connectivity to medial and lateral portions of anterior VTC, ii) show negative BOLD responses during visual perception with clear preferences for scenes and faces and iii) are differentially recruited during memory recall for either specific places or specific people. Further, at the whole-brain level we identify a second pair of anterior regions, which exhibit the same selective recruitment during memory recall for places and people, revealing a total of four subdivisions within MPC. Taken together, these findings provide converging evidence that the functional organization defining the medial-lateral axis of VTC is reflected along the ventral/posterior-dorsal/anterior axis of MPC, but in the context of memory recall.

## The functional organization of medial parietal cortex

The selective recruitment of MPC during recall of places or people is consistent with a diverse literature linking MPC with multiple memory processes (*Vilberg and Rugg, 2008*; *Wagner et al., 2005*; *Gilmore et al., 2015*; *Kim, 2013*; *McDermott et al., 1999*; *Gordon et al., 2017*). Unlike previous neuropsychological work (*Valenstein et al., 1987*; *Arnott et al., 2008*; *Gainotti et al., 1998*), which

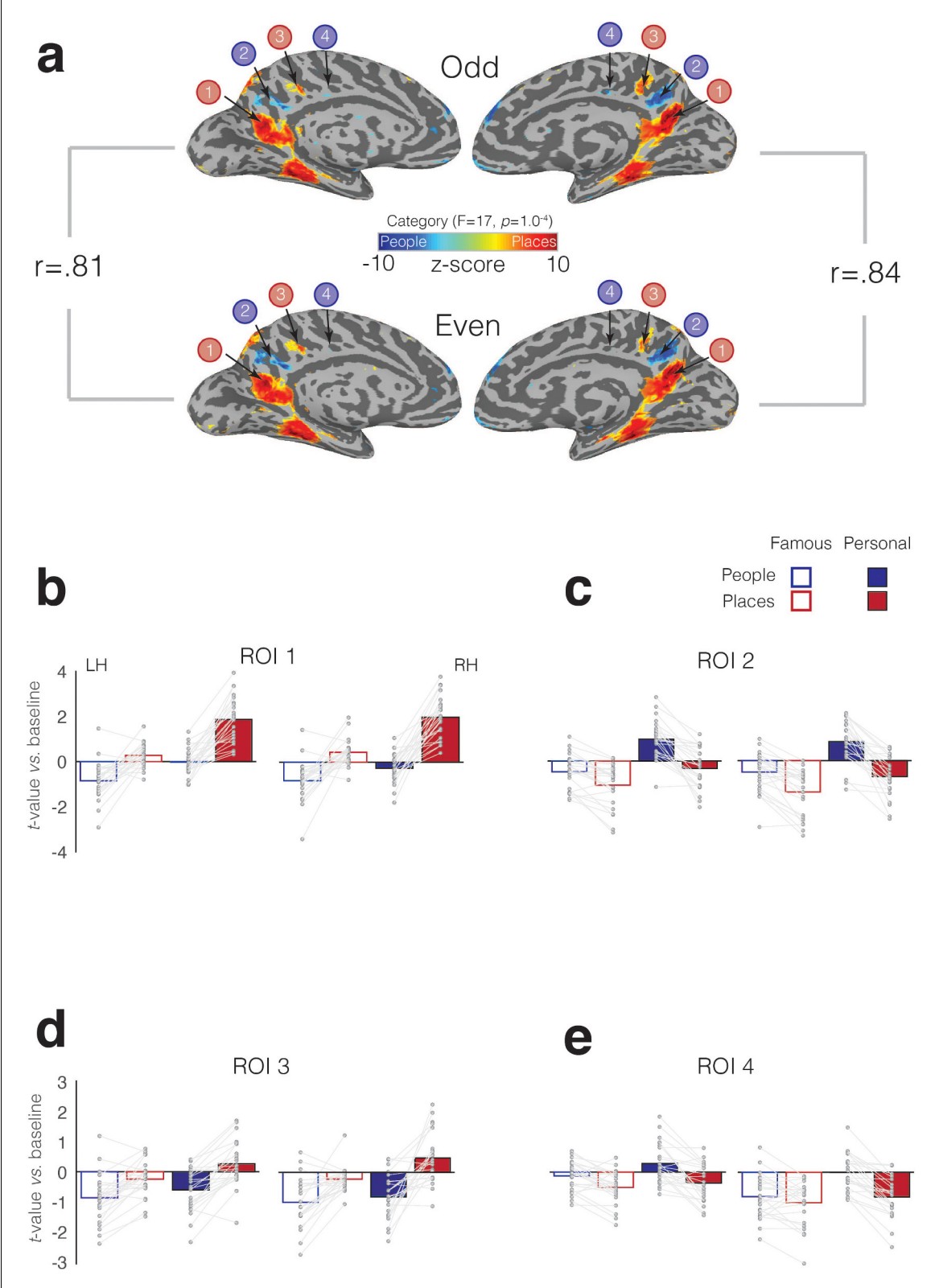

**Figure 7.** Split-half analysis and memory recall effects in four MPC subdivisions. (a) The effect of Category (node-wise p=1.0$^{-4}$, q = 5.8$^{-4}$) is overlaid onto medial views of both the left and right hemispheres for both independent halves of the data separately (Odd – top row, Even – bottom row). Cold colors represent regions of the brain more active during the recall of people, whereas hot colors represent regions of the brain more active during recall of places. Despite having half the amount of data, the alternating pattern of category-selectivity within MPC is present in both halves and

*Figure 7 continued on next page*

*Figure 7 continued*

hemispheres, respectively. The magnitude of the category effect (F-stat) was highly correlated across splits. The reported rho-values correspond to the correlation of the node-wise F-statistic for the effect of category in each hemisphere across the two splits. (B) Bars represent the mean response magnitude for each condition (*t*-value versus baseline) from ROI 1. Single participant data points are shown and are connected for each participant. (c-e) same as (a), but for ROIs 2–4.

DOI: https://doi.org/10.7554/eLife.47391.015

The following source data and figure supplements are available for figure 7:

**Source data 1.** Split-half analysis and memory recall effects in four MPC subdivisions.

DOI: https://doi.org/10.7554/eLife.47391.018

**Figure supplement 1.** Functional connectivity between MPC subdivisions and foveal/peripheral early visual cortex.

DOI: https://doi.org/10.7554/eLife.47391.016

**Figure supplement 1—source data 1.** Functional connectiviy values between MPC subdivisions and foveal/peripheral early visual cortex.

DOI: https://doi.org/10.7554/eLife.47391.017

lacked the spatial specificity to examine the heterogeneity of MPC or neuroimaging work that has associated MPC divisions with broad domain-level processes (*Wagner et al., 2005*; *Andrews-Hanna et al., 2010*), we provide evidence for a set of functionally dissociable sub regions along the ventral/posterior-dorsal/anterior axis of MPC that appear selective for places and people, reminiscent of category-selective areas along the medial-lateral axis of VTC.

Initially, we focused on two MPC subdivisions, defined on the basis of preferential functional connectivity to medial and lateral portions of anterior VTC, respectively. These subdivisions exhibited different (albeit negative) responses to visually presented categories and were differentially recruited during memory recall. During the recall of specific places, MPCv showed strong positive evoked responses, which contrasted with negative evoked responses during the recall of specific people. In contrast, MPCd showed largely the opposite pattern - large positive evoked responses during recall of personally familiar people, but not during the recall of either famous people or places. These data show a clear division between ventral/posterior and dorsal/anterior portions of MPC based on the content of the recalled memory. This division is broadly consistent with prior anatomical (Vogt, 2009; *Parvizi et al., 2006*) and functional imaging (*Margulies et al., 2009*; *Vidaurre et al., 2018*; *Foster and Parvizi, 2012*; *Margulies et al., 2016*; *Peer et al., 2015*) work that also identified divisions within MPC along this axis, but did not do so on the basis of recalled content.

The cortical locations of MPCv/MPCd are consistent with previous observations of memory related activity (*Andrews-Hanna et al., 2010*; *Peer et al., 2015*; *Gilmore et al., 2018*; *Kuhl and Chun, 2014*; *Chen et al., 2017*). Indeed, the locations of MPCv/MPCd are qualitatively similar to regions recruited when participant's mentally oriented by either making egocentric distance judgments between two places (i.e. which location is physically closest to them), or between two people (i.e. which of two people are personally closer to them), respectively (*Peer et al., 2015*). Although such a distance judgement undoubtedly requires recalling specific information, our findings suggest that an explicit distance task is not required to functionally isolate MPCv and MPCd. In the current study, participants were not asked to make social and spatial distance judgments, but merely to recall from memory either specific people or places. This, we believe, provides strong evidence for the role that MPCv/MPCd play in our ability to recall specific features from memory—and seemingly in a manner that recapitulates VTC organization.

Importantly, we also identify a second pair of anterior regions along the same ventral/posterior-dorsal/anterior axis, that fall anterior of the 'place' and 'people' regions reported by *Peer et al. (2015)*, revealing a total of four subdivisions selectively recruited during memory recall for people or places in an alternating pattern. One intriguing feature of this anterior pair of regions was that the pattern of selective recruitment remained despite an overall reduction in the magnitude of responses to recalled stimuli, as compared to MPCv/MPCd. The location of this anterior pair appears qualitatively similar to the 'time' region identified by *Peer et al. (2015)*. It is possible that the posterior and anterior pairs we identified play complementary and yet different roles in representing information about people and places. Elucidating any potential differences between these pairs of regions and how they may relate to the representations of time reported by *Peer et al. (2015)* are key goals for future work.

A major contribution of the current work is the demonstration that the functional organization defining the medial-lateral axis of VTC is reflected along the ventral/posterior-dorsal/anterior axis of MPC. The present work informs other recent parcellations of MPC (*Chrastil, 2018*); Power et al. 2014; *Braga and Buckner, 2017*; *Yeo et al., 2011*). For example, a recent meta-analysis attempted to divide a ventral portion of MPC (referred to as retrosplenial complex) on the basis of different fMRI task activations (*Chrastil, 2018*). Although the majority of MPC was found to be recruited during memory tasks, a ventral/posterior region was more likely to be recruited during scene and navigation-related tasks, whilst more dorsal/anterior portions were more likely to be involved in theory-of-mind and social/emotional tasks. Similarly, a recent report (*Andrews-Hanna et al., 2010*) separated social regions of MPC (i.e. those engaged with theory-of-mind tasks) from those involved with constructive memory or the formation of contextual associations (*Hassabis et al., 2007*; *Bar and Aminoff, 2003*). These divisions align well with the differential memory recall effects reported here. The dorsal and ventral divisions also align well with recent studies that have attempted to map social network representations (*Parkinson et al., 2017*) and representations of familiar scenes (*Sugiura et al., 2005*), respectively. Our results provide a framework for understanding these previously-reported mnemonic effects by highlighting the apparent organizational link between the ventral/posterior-dorsal/anterior axis of MPC and the medial-lateral axis of VTC. Perhaps more importantly, these robust effects were evoked by the relatively simple task of recalling items that were cued by word stimuli only (or even merely perceiving presented stimuli), as opposed to performing more complex contextual association (*Hassabis et al., 2007*), navigation (*Chrastil, 2018*), social judgment (*Parkinson et al., 2017*), or mental orientation tasks (*Peer et al., 2015*). Such strong recruitment of MPC through a simple paradigm is consistent with recent work showing that MPC (and the larger DMN) plays a role in simple spatial judgments of shapes and objects, particularly when made from memory (*Konishi et al., 2015*; *Murphy et al., 2018*; *Murphy et al., 2019*). Consequently, these simple paradigms pave the way for future research to potentially address functional heterogeneity in MPC using tasks that target specific cognitive processes.

## Relating MPC to large-scale cortical networks

The role of MPC is often considered in the context of the DMN (*Andrews-Hanna et al., 2010*). Although initially conceived as a singular entity (*Raichle et al., 2001*), the DMN has more recently been divided into two (*Shirer et al., 2012*) or three (*Andrews-Hanna et al., 2010*) subnetworks. In the 'three network' framework, much of MPC and ventral medial prefrontal cortex act as a 'core' that flexibly integrates information between the 'dorsal' and 'ventral' subnetworks (*Andrews-Hanna et al., 2010*). Our results challenge this 'core' conceptualization by suggesting that MPC is fractionated along the same lines as these subnetworks: the dorsal component of the DMN, which overlapped regions recruited during people memory, is associated with social/semantic processing (*Andrews-Hanna et al., 2010*) (*Figure 8a*), while, the ventral component, which is often referred to as a 'scene construction' or 'contextual association' (*Hassabis et al., 2007*; *Bar and Aminoff, 2003*) network overlaps with regions recruited during place memory (*Figure 8b*).

In the 'two network' framework (*Shirer et al., 2012*) the DMN is comprised of dorsal and ventral subnetworks without a clear integrative 'core'. This framework is also consistent with recent work using highly-sampled participants to identify two parallel and interdigitated networks spanning cortex (*Braga and Buckner, 2017*). These two functional networks—both of which appeared to overlap with the canonical DMN—could be differentiated through functional connectivity, given sufficient data, but stopped short of describing their specific functional organization. Our data provide a possible functional account of these networks by anchoring them to differential recruitment based on the content of perceived images as well as recalled memories, and moreover, demonstrate that such an interdigitated pattern of regions can be identified at the group-level given appropriate tasks.

The MPC regions we report also appear to overlap with the recently proposed 'posterior medial' (PM) memory system and the effects we observe are broadly consistent with the notion that the PM system is involved in recollecting episodic details, constructive uses of memory, and social cognition (*Ranganath and Ritchey, 2012*). However, the present work suggests a differentiation within the PM system based on the recall of places and people, which was not discussed originally, suggesting that MPC's role in memory recall does not fit neatly into a simple 'binary systems' model. More recently, a parietal memory network (PMN) (*Gilmore et al., 2015*; *McDermott et al., 2017*) has been identified, which includes regions within both lateral parietal cortex and MPC that are thought to be

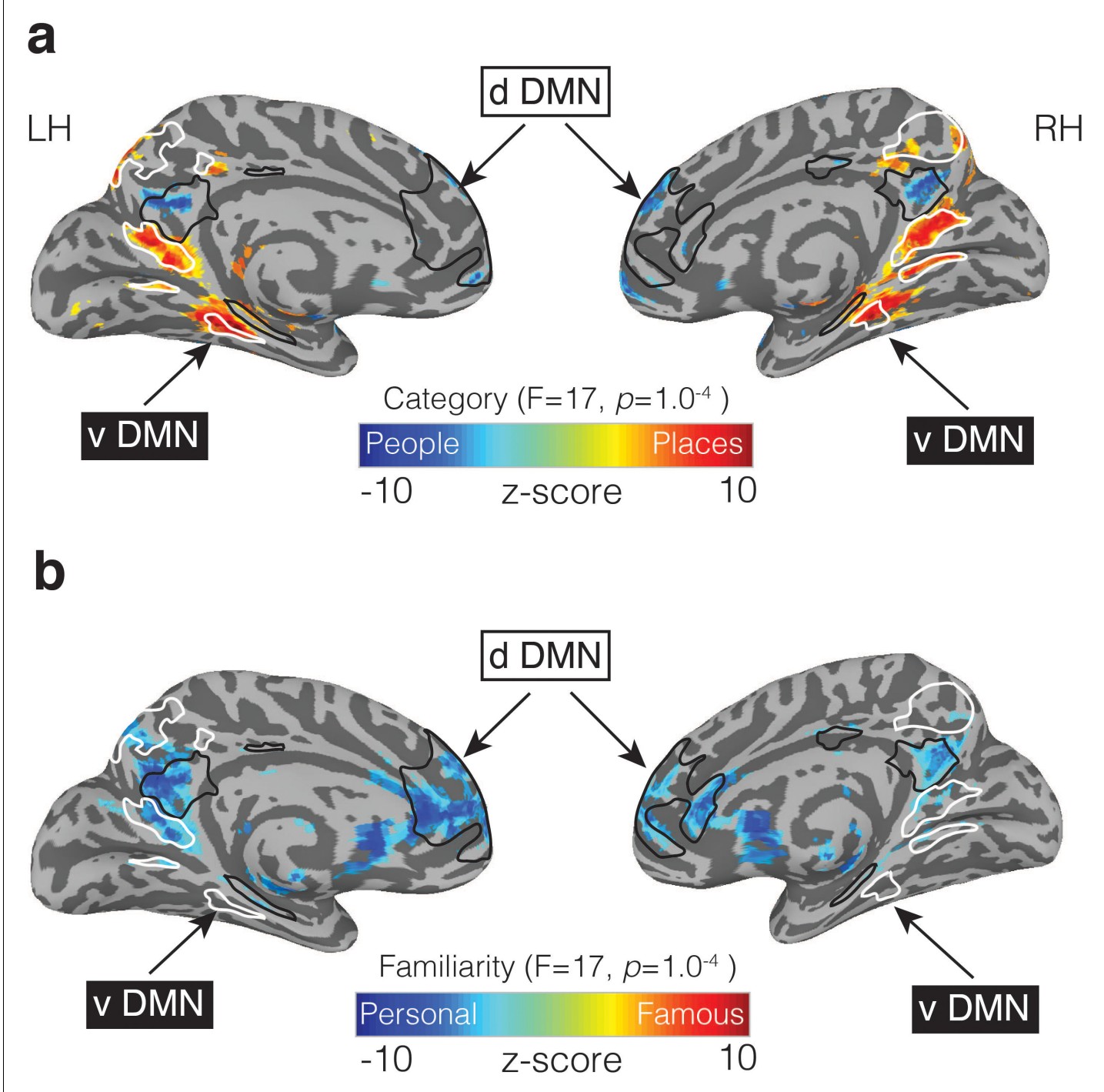

**Figure 8.** People and Place memory areas of MPC correspond to the dorsal and ventral DMN subnetworks. (**a**) Medial views of both the left and right hemispheres are shown (TT n27 surface). Overlaid onto these surfaces is the effect of Category (p=1.0$^{-4}$, q = 5.8$^{-4}$). Cold colors represent regions of the brain more active during the recall of people (famous and personal), whereas hot colors represent regions of the brain more active during recall of places (famous and personal). Masks of the dorsal DMN (dDMN) are outlined in black and show a striking spatial correspondence to regions more active during recall of people. In contrast, masks of the ventral DMN (vDMN) are outlined in white and correspond to regions more active during recall of places. DMN masks were taken from 38. (**b**) The effect of Familiarity is overlaid onto the same surfaces (p=1.0$^{-4}$, q = 5.8$^{-4}$). Cold colors represent regions of the brain more active during the recall of personally familiar stimuli (places and people), whereas hot colors represent regions of the brain more active during the recall of famous stimuli (places or people). Unlike in (**a**), regions showing a Familiarity effect overlap with regions from both the dDMN and vDMN.

DOI: https://doi.org/10.7554/eLife.47391.019

distinct from those within the DMN/PM (*Yang et al., 2014*; *Hu et al., 2019*; *Gilmore et al., 2019a*; *Gilmore et al., 2019b*) system. The memory effects we report overlap with DMN regions and appear close to, but separate from, those within the PMN (i.e., do not extend to mid-cingulate cortex or posterior aspects of the precuneus). This adjacency reinforces the conceptual separation of processes associated with the PMN and the DMN/PM system, respectively.

## Functional correspondence between VTC and MPC

Multiple interrelated functional organizations are thought to be represented along the medial-lateral axis of VTC (*Kanwisher et al., 1997*; *Deen et al., 2017*; *Levy et al., 2001*; *Arcaro and Livingstone, 2017*; *Konkle and Caramazza, 2013*). How the current data fit in the context of these organizations thus requires careful consideration.

Category-selectivity is one organizing principle thought to define the medial-lateral axis of VTC. Even at early stages of cortical development (*Deen et al., 2017*; *Arcaro and Livingstone, 2017*), category-selective regions appear in stereotypical locations across individuals. This consistency across individuals and developmental time-frames supports the notion that categories are represented within discrete regions of VTC. The pattern of functional-connectivity, perceptual and memory recall effects we report are consistent with the presence of this motif within MPC. However, other functional dimensions are also thought to be represented along the medial-lateral axis of VTC, and the extent to which they interact and relate to one-another, and in-turn relate to the organization of MPC, is an open question (*Arcaro and Livingstone, 2017*; *Livingstone et al., 2019*).

For example, eccentricity also varies systematically along the medial-lateral axis of VTC (*Levy et al., 2001*). Indeed, these eccentricity representations are so highly correlated with category preference (e.g. PPA is peripherally biased, whereas FFA is foveally biased) that eccentricity has been suggested to form a prototypical organization onto which such selectivity later develops with visual experience (*Silson et al., 2015*; *Arcaro and Livingstone, 2017*). It may be that the selective recruitment we observe in MPC during memory recall arises in a similar manner: That is, it is possible that the development of categorical-preference in VTC during perception drives the development of categorical-preference in MPC for memory. Such a proposition is compatible with recent work focusing on large-scale cortical gradients that conceptualized the DMN, including MPC, as an abstract extension of VTC (*Margulies et al., 2016*), although in the current study we observed effects that appeared to follow a distinct regional—rather than gradient-like—organization. Alternatively, it is possible that the manner in which mnemonic representations of places and people are encoded and retrieved in MPC, differ in a way reminiscent of how the visual perception of places and people in VTC differ in terms of peripheral/foveal stimulation (*Figure 7—figure supplement 1*). The correspondence between MPC organization and the multiple dimensions thought to be represented across VTC, are key questions for future research. Importantly, these accounts are consistent with theories that suggest the organization of category representations in the brain are determined by the underlying structural (i.e. anatomical) and functional (e.g. perceptual) template (*Martin, 2016*).

## Nature of responses within MPC during perception

The responses of MPCv/MPCd during perception share similarities with VTC but differ in important ways. For instance, responses in VTC during perception are characterized by larger evoked positive responses to stimuli of the preferred category (*Silson et al., 2016*). In contrast, perceptual responses within MPCv/MPCd were best characterized by negative evoked responses that were attenuated, although not extinguished, by category-preference. Despite appearing to share category representations, there was no clear relationship between the positively and negatively evoked responses between the VTC regions and their paired MPC region (e.g. anterior medial VTC – MPCv), at least at the across participant level. Although the relationship between negative BOLD-signal changes and the underlying neural activity is an area of ongoing research (*Shmuel et al., 2002*; *Shmuel et al., 2006*) and has been associated with inhibition in visual cortex (*Smith et al., 2004*), this response is consistent with the widely-observed 'task-negative' activation of the larger DMN and the MPC component of it (*Andrews-Hanna et al., 2010*). The current study did not compare directly the responses to perceived and subsequently recalled stimuli. It is possible that the degree of attenuation of the negative response during perception is related to the degree of recruitment during recall in MPCv/MPCd (for related discussion, see *Daselaar et al., 2004*; *Daselaar et al., 2009*).

## Conclusion

In this study we identified a consistent differentiation of regions within MPC, providing a new framework for understanding and investigating the functional organization of MPC and its role in memory retrieval. This differentiation was present at rest, in response to visually presented stimuli and finally through memory recall. Such division of MPC is consistent with previous anatomical (Vogt 2009) and functional (*Margulies et al., 2016*; *Peer et al., 2015*; *Foster and Parvizi, 2012*; *Andrews-Hanna et al., 2014*; *Leech et al., 2011*) distinctions but suggests that these divisions can be seen on the basis of recalled content. These data provide converging evidence that the functional organization defining the medial-lateral axis of VTC is reflected along the ventral/posterior-dorsal/anterior axis of MPC, but in the context of memory recall.

# Materials and methods

## Participants

Participants for all experiments were recruited from the DC area and NIH community. All participants were right-handed with normal or corrected-to-normal vision and neurologically healthy. All participants gave written informed consent according to procedures approved by the NIH Institutional Review Board (protocol 93 M-0170, clinical trials # NCT00001360). Participants were monetarily compensated for their time.

*Resting-state functional connectivity experiment:* Sixty-five participants (40 female), mean age = 24.67 ± 3.2 years) completed the resting-state functional connectivity experiment.

*Six category functional Localizer experiment:* Twenty-nine participants (21 female, mean age = 24.2 years) completed the functional localizer experiment.

*Memory experiment:* Twenty-four participants (17 female, mean age = 24.2 years) completed the memory experiment. The sample size for the memory experiment was based on prior work from our group (*Silson et al., 2019*) employing a very similar paradigm.

## Stimuli and tasks

*Six category functional localizer experiment*: Participants completed six functional localizer runs. During each run, color images from six stimulus categories (Scenes, Faces, Bodies, Buildings, Objects and Scrambled Objects) were presented at fixation (5 × 5° of visual angle) in 16 s blocks (20 images per block [300 ms per image, 500 ms blank]). Each category was presented twice per run, with the order of presentation counterbalanced across participants and runs. Participants responded via MRI compatible button box whenever the same image appeared sequentially. Stimuli for this and the other in-scanner tasks were presented using PsychoPy software (*Peirce, 2007*) (RRID:SCR_006571) from a Macbook Pro laptop (Apple Systems, Cupertino, CA).

*Memory Experiment:* Stimuli consisted of written names and words: 36 famous people, 36 famous places, 36 personally familiar people and 36 personally familiar places. The stimuli were provided by participants through a survey completed prior to the fMRI scan. Participants selected 36 known famous people and famous places from a list of 60 possible famous people (e.g., Tom Hanks, Angelina Jolie) and 92 possible famous places (e.g., Eiffel Tower, Times Square), and also provided the experimenters with the names of 36 people and 36 places that were personally familiar to them. Stimuli were presented in white 18-point Arial, all capital type against a black background.

During the task, on each trial participants were instructed to visualize the presented stimulus from memory as vividly as possible for the duration of the trial (10 s). Trials were separated by a variable inter-trial interval (2.5–7.5 s). In each of the six runs, there were six trials of each condition (famous people, famous places, personally familiar people and personally familiar places) presented in a randomized order, for a total of 24 trials per run (144 trials total).

*Post scan questionnaire:* After the scans were complete, participants completed a questionnaire in which they rated how vividly they were able to visualize from memory each stimulus presented during the memory runs. The stimuli were listed in the same order they appeared during the Memory Experiment and were rated on a 4-point Likert type scale (1 = not at all vivid; 4 = extremely vivid). If the participant could not visualize the stimulus at all while in the scanner, they checked a box on the questionnaire.

## Functional imaging parameters

Below we outline the imaging parameters for the three separate imaging experiments included in the current manuscript. All scans were performed on a 3.0T GE 750MRI scanner using a 32-channel head coil.

### Resting-state functional connectivity

All functional images were acquired using a BOLD-contrast sensitive three-echo echo-planar sequence (ASSET acceleration factor = 2, TEs = 14.9, 28.4, 41.9 ms, flip-angle = 65˚, bandwidth = 250.000 kHz, FOV = 24×24 cm, acquisition matrix = 64×64, resolution = 3.4×3.4 x 3.4 mm, slice gap = 0.3 mm, 34 slices per volume covering the whole brain). Respiratory and cardiac traces were recorded. Resting state scans lasted 21 min. The first 30 volumes were discarded to control for the state of arousal during the initial stages of data acquisition, leaving 20 min (600 volumes) for resting state functional connectivity analysis. This procedure has been used in other studies where long-duration resting state runs were collected.

### Six category functional localizer

All functional images were acquired using a BOLD-contrast sensitive standard EPI sequence (TE = 30 ms, TR = 2 s, flip-angle = 65 degrees, FOV = 192 mm, acquisition matrix = 64×64, resolution 2 × 2×2 mm, slice gap = 0.2 mm, 37 slices covering the occipital and temporal lobes.

### Memory experiment

All functional images were acquired using a BOLD-contrast sensitive three-echo echo-planar sequence (ASSET acceleration factor = 2, TEs = 12.5, 27.7, and 42.9 ms, flip angle = 75˚, 64 × 64 matrix, in-plane resolution = 3.2×3.2 mm, slice thickness = 3.5 mm). Repetition times (TRs) and acquired slices varied across different task conditions to be consistent with relevant prior work for each task. The memory task used a 2500 ms TR, with 35 slices collected. All slices were collected obliquely and were manually aligned to the AC-PC axis.

### fMRI data preprocessing

Data were preprocessed using AFNI (*Cox, 1996*) (RRID: SCR_005927) for all experiments. Below we outline the preprocessing steps taken during each experiment.

### Resting-state and memory experiments

The first 4 frames of each run were discarded to allow for T1 equilibration effects. Initial preprocessing steps for fMRI data were carried out on each echo separately. Slice-time correction was applied (3dTShift) and signal outliers were attenuated (3dDespike). Motion-correction parameters were estimated from the middle echo based on rigid-body registration of each volume to the first volume of the scan; these alignment parameters were then applied to the first and third echo. Data from all three acquired echoes were then registered to each participants' T1 image and combined to remove additional thermal and physiological noise using multi-echo independent components analysis (*Kundu et al., 2012*; *Kundu et al., 2013*). This procedure initially uses a weighted-average of the three echo times for each scan run to reduce thermal noise within each voxel. It subsequently performs a spatial ICA to separate time series components and uses the known properties of $T_2^*$ signal decay to separate putatively neuronal BOLD components from putative noise components. This is accomplished by comparing each component to a model that assumes a temporal dependence in signal decay (i.e., that is 'BOLD-like') and to a different model that assumes temporal independence (i.e., that is 'non-BOLD-like'). Components with a strong fit to the former and a poor fit to the latter model are retained for subsequent analysis. This procedure was conducted using default options in AFNI's tedana.py function. ME-ICA processed data from each scan were then aligned across runs for each participant.

### Six category functional localizer experiment

All images were motion corrected to the first image of the first run (3dVolreg), after removal of the appropriate 'dummy' volumes (8) to allow stabilization of the magnetic field. Following motion

correction, images were spatially smoothed (3dmerge) using a 5 mm full-width-half-maximum smoothing kernel.

## fMRI data analysis

### Resting-state functional connectivity winner-take-all analysis

Each participant's resting-state functional connectivity time series was aligned to the standard surface using 3dVol2Surf. We sought to assess the RSFC between VTC and MPC as a function of both the posterior-anterior and medial-lateral axes. The region of VTC we considered was longer in the posterior-anterior direction than it was wide in the medial-lateral direction and so we chose to sample it using a 3 × 2 ROI scheme. Accordingly, Six ROIs were defined that divided VTC along both the posterior-anterior and medial-lateral axes with respect to the mid-fusiform sulcus on a standard surface mesh that was aligned to the anatomy of an independent participant (not included in this study). The surface vertices within these ROIs were transformed into each individual participant's surface. This is standard for surface-based ROI analyses in AFNI (*Saad and Reynolds, 2012*).

The surface mesh used within SUMA contains a standard number of vertices, whilst also respecting an individual participant's specific gyral and sulcal pattern. Thus, drawing these regions based on the medial and lateral fusiform gyrus on the surface mesh provides highly accurate mapping of this anatomical landmark across participants (*Saad and Reynolds, 2012*).

For each participant, time series from these six ROIs were first extracted then scaled by the mean, before the unique connectivity of each parcel to the rest of the brain was calculated using multiple-regression. The 'winning' parcel at each node was then determined by the maximum beta-value for each parcel (e.g. anterior medial VTC), and the selectivity index of the node was determined by subtracting the mean beta-values of all other parcels from the winning parcel (e.g. selectivity index = anterior medial VTC – (middle medial VTC + posterior medial VTC + anterior lateral VTC + middle lateral VTC + posterior lateral VTC).

An alternative approach is to normalize the variance (z-score) in the time-series prior to running the winner-take-all analysis. This analysis produced qualitatively similar results to our original analysis and thus, we elected to keep our original definitions of MPCv/MPCd.

### Six category functional localizer analysis

A general linear model (GLM) approach was also used to analyze the functional localizer data. Specifically, a response model was built by convolving a standard gamma function with a 16 s square wave for each condition and compared against the activation time courses using Generalized Least Square (GLSQ) regression. Motion parameters and four polynomials accounting for slow drifts were included as regressors of no interest. To derive the response magnitude per condition, *t*-tests were performed between the condition-specific beta estimates (normalized by the grand mean of each voxel for each run) and baseline.

### Memory analysis

Analyses were conducted using a general linear model (GLM) and the AFNI programs 3dDeconvolve and 3dREMLfit. The data at each time point were treated as the sum of all effects thought to be present at that time point and the time series was compared against GLSQ model fit with REML estimation of the temporal auto-correlation structure. Responses were modeled by convolving a standard gamma function with a 10 s square wave for each condition of interest (Famous People, Famous Places, Personal People and Personal Places). Estimated motion parameters were included as additional regressors of non-interest, and fourth-order polynomials were included to account for slow drifts in the MR signal over time. Significance was determined by comparing the beta estimates for each condition (normalized by the grand mean of each voxel for each run) against baseline.

### Sampling of data to the cortical surface

In each participant, the analyzed functional data were projected onto surface reconstructions of each individual participant's hemispheres using the Surface Mapping with AFNI (SUMA) software. First, data were aligned to high-resolution anatomical scans (align_epi_anat.py). Once aligned, these data were projected onto the cortical surface (3dVol2Surf) and smoothed by a 2 mm FWHM Gaussian kernel.

### Linear mixed effects analysis

To look at the whole brain memory effects we employed a linear-mixed-effects model (3dLME) in each hemisphere separately. The model comprised two factors: Category (Places, People) and Familiarity (Famous, Personal). At the whole brain-level, we did not observe any significant interactions, but both robust main effects were significant.

### Split-half analysis

For each participant, the six memory runs were divided into Odd and Even splits (three runs each). In each split, we performed the same LME. At the whole brain-level, we did not observe any significant interactions, but both robust main effects were significant. Throughout MPC we observed 4 ROIs that showed an alternating pattern of category-selective recall in both splits. To quantify these effects, we first defined each region within a split (e.g. Odd) and then sampled the data from the other half (e.g. Even). To avoid any potential bias in node selection, this process was reversed, and the average computed.

## Statistical approach

Statistical analyses of both behavioral and functional data were performed using the SPSS software package (IBM). For all analyses we conducted repeated measures ANOVAs. When a significant three-way interaction was present, we performed two separate two-way ANOVAs to explore the nature of the interaction.

## Acknowledgements

We thank members of the Laboratory of Brain and Cognition for helpful comments on earlier version of the manuscript. This research was supported by the Intramural Research Program (ZIAMH002909) of the National Institutes of Health - National Institute of Mental Health Clinical Study Protocol 93 M-0170, NCT00001360.

## Additional information

### Funding

| Funder | Grant reference number | Author |
| --- | --- | --- |
| National Institutes of Health | ZIAMH002909 | Edward H Silson<br>Adam Steel<br>Alexis Kidder<br>Adrian W Gilmore<br>Chris I Baker |

The funders had no role in study design, data collection and interpretation, or the decision to submit the work for publication.

### Author contributions

Edward H Silson, Adam Steel, Conceptualization, Data curation, Formal analysis, Visualization, Writing—original draft, Writing—review and editing; Alexis Kidder, Data curation, Visualization, Writing—original draft, Writing—review and editing; Adrian W Gilmore, Chris I Baker, Conceptualization, Writing—original draft, Writing—review and editing

### Author ORCIDs

Edward H Silson https://orcid.org/0000-0002-6149-7423
Adam Steel http://orcid.org/0000-0001-8876-933X
Adrian W Gilmore http://orcid.org/0000-0001-8910-5009
Chris I Baker http://orcid.org/0000-0001-6861-8964

## Ethics

Human subjects: All participants gave written informed consent according to procedures approved by the NIH Institutional Review Board (protocol 93-M-0170, clinical trials # NCT00001360).

## Decision letter and Author response

Decision letter https://doi.org/10.7554/eLife.47391.023
Author response https://doi.org/10.7554/eLife.47391.024

---

# Additional files

## Supplementary files

• Supplementary file 1. Full statistical breakdown of behavioral and functional data. Supplementary file 1a: Statistical analysis of behavioral responses collected directly after the memory experiment. Data are provided for both subjective vividness and the proportion of missed trials. Table includes F-values, degrees of freedom (*df*), *p*-values and estimates of effect size using partial eta squared. In each case, a two-way repeated measures ANOVA was conducted with Category (Places, People) and Familiarity (Famous, Personal) as within-participant factors. In the case of vividness ratings, both main effects of Category and Familiarity were significant, reflecting on average higher vividness ratings for the recall of people over places and for personal over famous stimuli. The significant Category by Familiarity interaction reflects a larger familiarity difference (Personal >Famous) during recall of places over people. For the proportion of missed trials, neither main effect was significant, but their interaction was. This interaction is driven by more missed trials for famous places than people, but fewer missed trials for personal scenes than people. Supplementary file 1b: Statistical analysis of memory effects in MPCv and MPCd. Table includes Fvalues, degrees of freedom (*df*), *p*-values and estimates of effect size using partial eta squared. MPCv showed significant main effects of Category, Familiarity and Hemisphere. These were qualified by a significant three-way interaction, reflecting a larger familiarity difference (Personal >Famous) between categories (Places > People) in the right over left hemisphere. MPCd also showed significant main effects but did not show a significant three-way interaction. Importantly, however, MPCd did show the predicted Category by Familiarity interaction, which reflects a larger familiarity difference for the recall of people over places. Supplementary file 1c: Statistical analysis of memory effects in ROI 1. Table includes Fvalues, degrees of freedom (*df*), *p*-values and estimates of effect size using partial eta squared. ROI one showed significant main effects of Category, Familiarity and Hemisphere. These were qualified by a significant three-way interaction, reflecting a larger familiarity difference (Personal >Famous) between categories (Places > People) in the right over left hemisphere. Supplementary file 1d: Statistical analysis of memory effects in ROI 2. Table includes Fvalues, degrees of freedom (*df*), *p*-values and estimates of effect size using partial eta squared. ROI two showed significant main effects of Category, Familiarity and Hemisphere, but did not show a significant three-way interaction. Importantly, however, ROI two did show the predicted Category by Familiarity interaction, which reflects a larger familiarity difference for the recall of people over places. Supplementary file 1e: Statistical analysis of memory effects in ROI 3. Table includes Fvalues, degrees of freedom (*df*), *p*-values and estimates of effect size using partial eta squared. ROI three showed significant main effects of Category, Familiarity, but not Hemisphere. Although ROI three did not show a significant three-way interaction, ROI three did show the predicted Category by Familiarity interaction, which reflects a larger familiarity difference for the recall of places over people. Supplementary file 1f: Statistical analysis of memory effects in ROI 4. Table includes Fvalues, degrees of freedom (*df*), *p*-values and estimates of effect size using partial eta squared. ROI four showed significant main effects of Category, Familiarity and Hemisphere. These were qualified by a significant three-way interaction, reflecting a larger familiarity difference (Personal >Famous) between categories (People > Places) in the right over left hemisphere. Supplementary file 1g: Statistical analysis of memory effects in PPA and FFA. Table includes Fvalues, degrees of freedom (*df*), *p*-values and estimates of effect size using partial eta squared. PPA showed significant main effects of Category and Familiarity, but not Hemisphere. PPA only showed a significant Category by Familiarity interaction, reflecting a larger familiarity difference (Personal >Famous) between categories (Places > People) with no clear difference between hemispheres. In contrast, FFA showed significant main effects of Category and Hemisphere, but not

Familiarity. These were qualified by a significant three-way interaction, which reflects the presence of category and familiarity in the left hemisphere, but only the effect of category in the right hemisphere. Supplementary file 1h: Statistical analysis of memory effects in the Hippocampus and Amygdala Table includes Fvalues, degrees of freedom (*df*), p-values and estimates of effect size using partial eta squared. The Hippocampus showed significant main effects of Category and Familiarity, but not Hemisphere. Only the Category by Hemisphere interaction was significant. The Amygdala showed only a significant effect of Category, with larger responses for the recall of people irrespective of the level of familiarity. Supplementary file 1i: Statistical analysis of the resting-state functional connectivity between MPCv, MPCd and Foveal, Peripheral portions of early visual cortex (EVC). Table includes F-values, degrees of freedom (*df*), p-values and estimates of effect size using partial eta squared. The main effects of ROI and Hemisphere were not significant, but the main effect of Eccentricity was reflecting on average stringer connectivity with peripheral than foveal portions of EVC. These were qualified however by a significant three-way interaction, reflecting on average stronger connectivity between MPCv and peripheral EVC, but stronger connectivity between MPCd and foveal EVC in the left over right hemispheres.
DOI: https://doi.org/10.7554/eLife.47391.020

• Transparent reporting form
DOI: https://doi.org/10.7554/eLife.47391.021

## Data availability

Source data files have been provided for Figures 4 and 7 and Supplementary Figures 2, 3, and 4.

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
