## [Decision Letter]

Thank you for submitting your article "Distinct subdivisions of human medial parietal cortex support recollection of people and places" for consideration by *eLife*. Your article has been reviewed by three peer reviewers, one of whom is a member of our Board of Reviewing Editors, and the evaluation has been overseen by Richard Ivry as the Senior Editor. The following individuals involved in review of your submission have agreed to reveal their identity: Robert Leech (Reviewer #2) and Jonathan Smallwood (Reviewer #3).

The reviewers have discussed the reviews with one another and the Reviewing Editor has drafted this decision to help you prepare a revised submission.

Summary:

Across three experiments, the authors demonstrate the division of human medial parietal cortex (MPC) into 4 alternating regions that are preferentially activated during memory recall of familiar people or familiar places. The first experiment shows that there are two MPC regions (MPCv and MPCd) that are respectively coupled (during the resting-state) with a medial anterior ventral temporal cortex (VTC) region and a lateral anterior VTC region (on either side of the mid-fusiform sulcus). The second experiment shows that MPCv is the least de-activated when subjects are visually presented with scenes, while MPCd is the least de-activated when subjects are visually presented with faces. The third experiment shows that MPCv was most strongly activated during memory recall of familiar places, while MPCd was most strongly activated during memory recall of familiar people. A whole brain analysis then revealed 2 more alternating patches in MPC that are strongly activated during memory recall of familiar places and familiar people respectively. The reviewers and editors felt that the experiments are well motivated and the analyzes are well-conducted and appropriate. The conclusions are also generally sound. Our main concern is that this work should be better situated with respect to the broader literature, some of which we have listed below under "essential revisions".

Essential revisions:

1) There is some important prior literature that is not cited and that as a result the Introduction is incomplete. For example, the authors state that previous literature "stopped short of describing a clear organizational structure. Thus, identifying a functional organization that provides synthesis across MPC is an important goal spanning multiple research fields…" However, the division of the posteriomedial surface of the cortex along dorsal-ventral/anterior-posterior gradients has been described in prior work e.g., cytoarchitecture and perfusion (e.g., Vogt, 2009), structural connectivity (Parvizi et al., 2006) functional connectivity (in humans and macaque monkeys, Margulies et al., 2009), MEG (Vidaurre et al., 2018) and ECOG (Foster et al., 2012) as well as other studies (e.g. Leech, et al., 2011, Bzdok et al., 2015, Peer et al., 2015). Other work has embedded the PCC as well as the rest of the default mode network in the context of large-scale cortical gradients, that include those in the ventral stream that this study is focused on (i.e. Margulies et al., 2016). In particular this viewpoint sees the default mode network as the most abstract extension of the ventral visual stream, and this has been shown to be important for memory retrieval (Murphy et al., 2018, 2019 both in Neuroimage). The Introduction should reflect this prior literature. Some description of the anatomical regions that compose the MPC would also be useful as would references to reviews other than e.g., Chrastil, which are focused only on the retrosplenial cortex (a relatively small portion of the human MPC).

2) This issue also applies to the Discussion e.g., "…demonstrate a more heterogenous organization within MPC than previously reported (Vilberg and Rugg, 2008; Wagner et al., 2005; Gilmore et al., 2015; Kim, 2013; Valensteinet et al., 1987; Arnott et al., 2008; Gainotti et al., 1998)."; subsection “Functional correspondence between VTC and MPC; "Perhaps more importantly, these robust effects were evoked by the relatively simple task of recalling items that were cued by word stimuli only (or even merely perceiving presented stimuli), as opposed to performing more complex contextual association (Hassabis et al., 2007), navigation (Chrastil, 2018) or social judgment tasks Parkinson et al., 2017)." Overall, the discussion and conclusions should reflect the wider work about the MPC's functional architecture: e.g., while the results suggest a functional dissociation between dorsal and ventral MPC, they might not really indicate the functional role of the dorsal region: e.g., task related BOLD signal changes from this region are small, especially given the high level of metabolism across much of the MPC).

3) Peer et al., 2015 demonstrated three MPC regions: one activated most strongly when orienting to space (the location is very similar to MPCv in this paper), one activated most strongly when orienting to person (the location is very similar to MPCd in this paper) and one activated most strongly when orienting in time. Thus, it's great that the authors' results are consistent with Peer and colleagues. However, the authors should also situate their work with respect to Peer and colleagues in the Introduction and/or Discussion.

---

## [Author Response]

Essential revisions:1) There is some important prior literature that is not cited and that as a result the Introduction is incomplete. For example, the authors state that previous literature "stopped short of describing a clear organizational structure. Thus, identifying a functional organization that provides synthesis across MPC is an important goal spanning multiple research fields…" However, the division of the posteriomedial surface of the cortex along dorsal-ventral/anterior-posterior gradients has been described in prior work e.g., cytoarchitecture and perfusion (e.g., Vogt, 2009), structural connectivity (Parvizi et al., 2006) functional connectivity (in humans and macaque monkeys, Margulies et al., 2009), MEG (Vidaurre et al., 2018) and ECOG (Foster et al., 2012) as well as other studies (e.g. Leech, et al., 2011, Bzdok et al., 2015, Peer et al., 2015). Other work has embedded the PCC as well as the rest of the default mode network in the context of large-scale cortical gradients, that include those in the ventral stream that this study is focused on (i.e. Margulies et al., 2016). In particular this viewpoint sees the default mode network as the most abstract extension of the ventral visual stream, and this has been shown to be important for memory retrieval (Murphy et al., 2018, 2019 both in Neuroimage). The Introduction should reflect this prior literature. Some description of the anatomical regions that compose the MPC would also be useful as would references to reviews other than e.g., Chrastil, which are focused only on the retrosplenial cortex (a relatively small portion of the human MPC).

We agree that a broader discussion of previous literature would better contextualize our findings. Accordingly, we have added sections to the Introduction describing both the anatomical boundaries subsumed by the MPC label, as well as, how previous work has identified divisions within MPC using both anatomical and multiple functional approaches. These sections are pasted below for reference and appear in blue in the revised manuscript.

“Human medial parietal cortex (MPC), a core component of the default mode network (DMN) (Andrews-Hanna et al., 2010), comprises a relatively large expanse of cortex, spanning the parieto-occipital sulcus to the splenium of the corpus collosum anteriorly and dorsally to include the precuneus and both the ventral and dorsal portions of the posterior cingulate cortex (Bzdok et al., 2015).”

“Network analyses using resting-state-functional-connectivity (RSFC) have identified either a single DMN “hub” region (Buckner et al., 2008) or multiple networks (Power et al., 2014; Braga and Buckner, 2017; Gilmore et al., 2018) that are often described as DMN subnetworks (Andrews-Hanna et al., 2010; Yeo et al., 2011; Doucet et al., 2011). […] Beyond MPC’s link with the DMN, others have described divisions of MPC along both the posterior-anterior and ventraldorsal axes in terms of cytoarchitecture (Vogt, 2009), structural connectivity (Parvi zi et al., 2006), RSFC (Marguiles et al., 2009; Vidaurre et al., 2018; Bzdok et al., 2015), and electrocorticography (Foster et al., 2012; Daitch and Parvizi, 2018).”

2) This issue also applies to the Discussion e.g., "…demonstrate a more heterogenous organization within MPC than previously reported (Vilberg and Rugg, 2008; Wagner et al., 2005; Gilmore et al., 2015; Kim, 2013; Valensteinet et al., 1987; Arnott et al., 2008; Gainotti et al., 1998)."; subsection “Functional correspondence between VTC and MPC; "Perhaps more importantly, these robust effects were evoked by the relatively simple task of recalling items that were cued by word stimuli only (or even merely perceiving presented stimuli), as opposed to performing more complex contextual association (Hassabis et al., 2007), navigation (Chrastil, 2018) or social judgment tasks Parkinson et al., 2017)." Overall, the discussion and conclusions should reflect the wider work about the MPC's functional architecture: e.g., while the results suggest a functional dissociation between dorsal and ventral MPC, they might not really indicate the functional role of the dorsal region: e.g., task related BOLD signal changes from this region are small, especially given the high level of metabolism across much of the MPC).

In the revised manuscript we now discuss how our results mesh with the existing literature. The added text is pasted below for reference.

**“**This division is broadly consistent with prior anatomical (Vogt, 2009; Parvizi et al., 2006) and functional imaging (Marguiles et al., 2009; Vidaurre et al., 2018; Foster et al. 2012; Marguiles et al., 2016; Peer et al., 2015) work that also identified divisions within MPC along this axis, but did not do so on the basis of recalled content.”

“The cortical locations of MPCv/MPCd are consistent with previous observations of memory related activity (Andrews-Hanna et al., 2010; Peer et al. 2015; Gilmore et al. 2018; Kuhl and Chun, 2014; Chen et al., 2017). […] Elucidating any potential differences between these pairs of regions and how they may relate to the representations of time reported by Peer et al., 2015, are key goals for future work.”

**“**Such strong recruitment of MPC through a simple paradigm is consistent with recent work showing that MPC (and the larger DMN) plays a role in simple spatial judgments of shapes and objects, particularly when made from memory (Konishi et al., 2015; Murphy et al., 2018; 2019). Consequently, these simple paradigms pave the way for future research to potentially address functional heterogeneity in MPC using tasks that target specific cognitive processes.”

“Such a proposition is compatible with recent work focusing on largescale cortical gradients that conceptualized the DMN, including MPC, as an abstract extension of VTC (Margulies et al., 2016), although in the current study we observed effects that appeared to follow a distinct regional—rather than gradient-like— organization.”

“The current study did not compare directly the responses to perceived and subsequently recalled stimuli. It is possible that the degree of attenuation of the negative response during perception is related to the degree of recruitment during recall in MPCv/MPCd (for related discussion, see Daselaar et al., 2004; 2009).”

“Such division of MPC is consistent with previous anatomical (Vogt, 2009) and functional (Marguiles et al., 2016; Peer et al., 2015; Foster et al., 2012, Andrews-Hanna et al., 2014; Leech et al., 2011) distinctions, but suggests that these divisions can be seen on the basis of recalled content.”

3) Peer et al., 2015 demonstrated three MPC regions: one activated most strongly when orienting to space (the location is very similar to MPCv in this paper), one activated most strongly when orienting to person (the location is very similar to MPCd in this paper) and one activated most strongly when orienting in time. Thus, it's great that the authors' results are consistent with Peer and colleagues. However, the authors should also situate their work with respect to Peer and colleagues in the Introduction and/or Discussion.

The Peer et al. results are indeed consistent with our own, and we agree that a broader discussion of the similarities and differences between the two studies is needed. The overall spatial location of the ‘space’ and ‘person’ regions shown by Peer and colleagues appear qualitatively to correspond with the MPCv and MPCd regions we identify. The major differences, however, are in how these regions were recruited. Peer and colleagues presented participants with two stimuli on each trial (either names of cities, names of people or events in time) and asked them to make an explicit egocentric distance judgement, whereas in the current paradigm participants were simply asked to recall from memory the target stimulus as vividly as possible. We believe our paradigm provides strong evidence that MPCv and MPCd play a general role in recalling from memory specific people and places and, moreover, that explicit distance judgements or decisions are not necessary to engage these regions. We now include a discussion of the similarities and differences between the two studies in the revised manuscript.

“The cortical locations of MPCv/MPCd are consistent with previous observations of memory related activity (Andrews-Hanna et al., 2010; Peer et al. 2015; Gilmore et al., 2018; Kuhl and Chun, 2014; Chen et al., 2017). […] Elucidating any potential differences between these pairs of regions and how they may relate to the representations of time reported by Peer et al., 2015, are key goals for future work.”